# Uncertainty-Guided Self-Learning Framework for Semi-Supervised Multi-Organ Segmentation

Natália Alves[1] and Bram de Wilde[1]

Diagnostic Image Analysis Group, Department of Medical Imaging, Radboud University Medical Center, 6500 HB Nijmegen, The Netherlands
{natalia.alves, bram.dewilde}@radboudumc.nl

**Abstract.** Automatic multi-organ segmentation in medical imaging has important clinical applications, but manual voxel-level annotations are time and labour-consuming, limiting the annotated data available for training. We propose an uncertainty-guided framework for multi-organ segmentation on CT scans that uses a small labelled dataset to leverage a large unlabelled dataset in a semi-supervised setting. First, we train five models to segment 13 abdominal organs using 50 manually labelled training cases and 5-fold cross-validation. Then, we use these models to generate pseudo-labels for 2000 unlabelled cases and estimate the uncertainty associated with the pseudo-labels by calculating the pair-wise Dice score (DSC) for the five individual predictions. Cases with pair-wise mean DSC>0.9 for all organs are included in the training set at the next iteration, together with the respective pseudo-labels. This process is repeated for four iterations. All selected cases are combined in the last iteration, and a single model is trained to reduce the computational costs associated with ensembling. The self-configuring method for biomedical image segmentation nnU-Net was used to train the segmentation models. We obtained a mean DSC of 0.8388 on the validation set with the network trained using the labelled data alone. The Dice score improved to 0.8874 in the final iteration of the model, trained with the 50 labelled cases and 1813 unlabelled cases with pseudo-labels. On the final test set, the mean DSC was 0.8685, and the mean inference time per case was 42 seconds. All code is open-source and available on GitHub (https://github.com/DIAGNijmegen/flare22-brananas).

**Keywords:** Semi-supervised learning · Uncertainty · Pseudo-labeling

## 1 Introduction

Abdominal organ segmentation on medical images has many important applications, such as organ quantification, surgical planning, and disease diagnosis. Recently, deep learning models, in particular convolutional neural networks (CNNs), have shown outstanding performance at abdominal organ segmentation across different datasets and image modalities [5]. CNNs require large-scale labelled datasets for model training to generalise well to unseen cohorts. However, the manual voxel-level annotation of multiple abdominal organs is time-

consuming and labour-intensive, limiting the number of samples available for training.

Semi-supervised learning (SSL) is a training strategy that uses a small amount of labelled data to leverage a large amount of unlabelled data [11]. In the medical domain, popular SSL techniques include transfer learning from a distant or related task, contrastive learning, and self-supervised representation learning with automatically generated labels [1]. A common approach for semi-supervised medical image segmentation is self-training, which consists of generating automatic pseudo-labels for the unlabelled data using the available annotated cases [1]. In particular, a segmentation CNN is first trained on the labelled samples, the trained model then segments the unlabelled samples, and finally, these samples, or a subset of these samples, are added to the training set. This process can be repeated several times, improving the pseudo-labels generated for the unlabelled data.

An inherent risk of the self-training method is the generation of low-quality pseudo-labels, which can lead to noisy training and poor generalisation. In order to lower this risk, only the most accurate pseudo-labels should be added to the training set at each iteration. This can be done, for example, by selecting the unlabelled cases with the most confident softmax output predictions. However, due to poor calibration of neural networks, wrong predictions can have high confidence, leading to the selection of low-quality pseudo-labels [9]. A more robust method also incorporates uncertainty quantification into the selection method. In a recent study, Rizve et al. propose an uncertainty-aware pseudo-label selection framework for natural image classification tasks, which improves pseudo-labelling accuracy by drastically reducing the amount of noise encountered in the training process [9].

In this work, we combine the state-of-the-art self-configuring method for biomedical image segmentation nnU-Net [5] with deep-ensemble uncertainty estimation [6] to build a self-learning framework that leverages prediction uncertainty to guide the pseudo-label selection procedure.

## 2 Method

### 2.1 Preprocessing

The proposed solution is based on the nnU-Net self-configuring framework for medical image segmentation [5]. All images are resampled to a resolution of $1.2 \times 1.2 \times 4$ mm$^3$. No other preprocessing is performed before training with the nnU-Net framework [5]. The framework calculates the global mean and standard deviation of the training dataset and applies z-score normalization to each instance, followed by clipping to the 0.5 and 99.5 percentiles. This framework applies its own cropping and intensity normalization strategies. The framework zero-pads patches to $40x224x192$ if needed.

## 2.2  Proposed Method

The network architecture is the 3D full resolution variant of the nnU-Net framework [5], which is illustrated in Fig. 1. The loss function is the sum of Dice and Cross Entropy loss, as employed by default by nnU-Net.

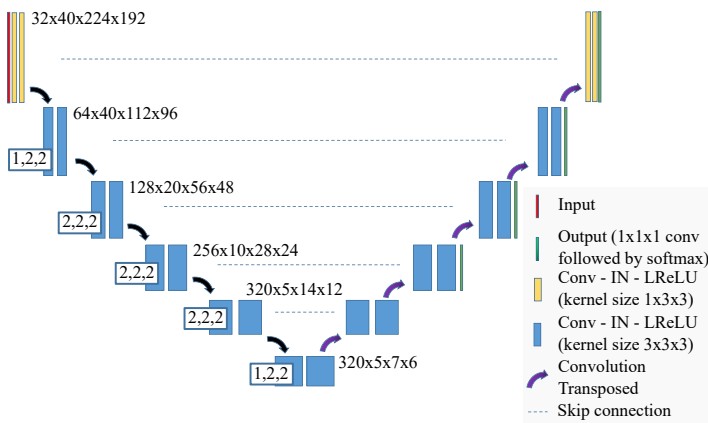

Fig. 1: 3D U-Net variant automatically selected by the nnU-Net framework, when applying it to the labelled data (see section 3.1) resampled to $1.2 \times 1.2 \times 4$ mm$^3$.

For training, we propose an iterative uncertainty-guided process to generate pseudo-labels for the 2000 unlabelled cases in the dataset. Initially, we train 5 models for 1000 epochs each with the 50 labelled cases using 5-fold cross validation. All models do individual inference on the 2000 unlabelled cases, which allows us to evaluate uncertainty for each case. The uncertainty metric we use is the mean pairwise DSC per organ considering the 5 individual models, $\overline{DSC}_i$, given by Equation (1). Here, $i$ is the organ index and $S_{ig}$ is a segmentation of organ $i$ by model $g$. Instead of using all 2000 cases for the next training iteration, we discard cases where the mean pairwise DSC is below 0.9 for at least one organ ($\min_i \overline{DSC}_i < 0.9$). In the next iteration, we train 5 models again using 5-fold cross validation, but now combining the 50 annotated cases with the pseudo-labelled cases that passed the uncertainty criterion. These pseudo-labels are the ensemble of the predictions generated by the 5 individual models. This process is continued for three iterations, after which we train a single model on all cases which pass the uncertainty threshold. The framework is depicted in Fig. 2.

$$\overline{DSC}_i = \frac{1}{10} \sum_{g=1}^{4} \sum_{h=g+1}^{5} DSC(S_{ig}, S_{ih}) \tag{1}$$

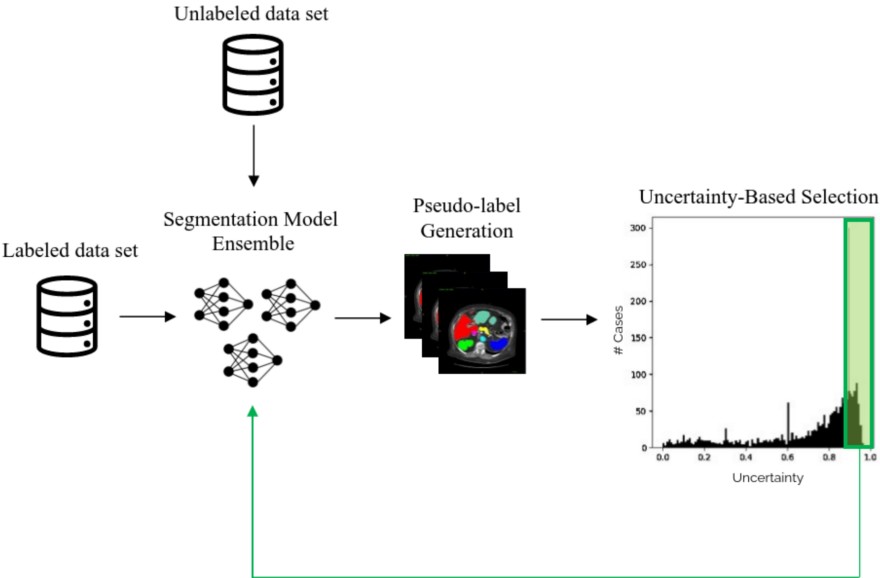

Fig. 2: Overview of the uncertainty-guided self-learning framework. A segmentation model ensemble is trained with a labeled data set. The ensemble generates pseudo-labels and using uncertainty all cases are ranked. A subset of cases which pass an uncertainty criterion (green rectangle) are used in a new training iteration, along with their pseudo-labels.

### 2.3   Post-processing

The inference is performed with a modified version of the nnU-Net framework. In particular, we increase the step size of strided inference from 0.5 to 0.9. This reduces the amount of 3D patches that have to pass through the network, reducing GPU time by a factor of 2. Additionally, we disable test-time augmentation to reduce GPU time by a factor of 2.5, as we observed modest gains in performance.

For post-processing, we start by correcting small noisy segmentations by removing all structures with a volume lower than the minimum volume for that organ in the labelled training set multiplied by 0.5. Then, we perform binary closing to the left and right adrenal gland masks with a $5 \times 5 \times 5$ kernel to connect any separate components on these small structures. We then mask the whole multi-organ segmentation based on the aorta, inferior vena cava and esophagus predictions to remove noisy labels outside the abdominal region. All slices above the first slice containing at least one of these structures and all slices below the last slice containing at least one of these structures are set to 0. We observed that lesions in the liver were often wrongly segmented as gallbladder, driving down the Dice for both structures. To correct this issue, we check every separate connected component segmented as gallbladder and if it is completely contained within the liver on any 2D slice (in the axial plane), we set the component to liver. Finally, to further remove smaller noisy structures, for every organ we remove any connected component with a volume smaller than 10 % of the whole segmented volume for that organ.

## 3   Experiments

### 3.1   Dataset and evaluation measures

The FLARE 2022 challenge is an extension of the FLARE 2021 [7] with more segmentation targets and more diverse abdomen CT scans. The dataset is curated from more than 20 medical groups under the license permission, including MSD [10], KiTS [3,4], AbdomenCT-1K [8], and TCIA [2]. The training set includes 50 labelled CT scans with pancreas disease and 2000 unlabelled CT scans with liver, kidney, spleen, or pancreas diseases. The validation set includes 50 CT scans with liver, kidney, spleen, or pancreas diseases. The testing set includes 200 CT scans where 100 cases have liver, kidney, spleen, or pancreas diseases and the other 100 cases have uterine corpus endometrial, urothelial bladder, stomach, sarcomas, or ovarian diseases. All the CT scans only have image information and the center information is not available.

The evaluation measures consist of two accuracy measures: Dice Similarity Coefficient (DSC) and Normalized Surface Dice (NSD), and three running efficiency measures: running time, area under GPU memory-time curve, and area under CPU utilization-time curve. All measures will be used to compute the ranking. Moreover, the GPU memory consumption has a 2 GB tolerance.

## 3.2   Implementation details

**Environment settings** The development environments and requirements are presented in Table 1.

Table 1: Development environments and requirements.

| | |
|---|---|
| Ubuntu version | 20.04 |
| CPU | Intel(R) Xeon(R) Gold 6152 CPU @ 2.10GHz (9 cores) |
| RAM | 30GB |
| GPU (number and type) | 1 NVIDIA RTX 2080 Ti |
| CUDA version | 11.3 with cudnn 8 |
| Programming language | Python 3.9 |
| Deep learning framework | PyTorch (Torch 1.11, torchvision 0.12, apex 0.1) |
| Specific dependencies | nnunet |

**Training protocols** The data augmentation strategy is determined automatically by the nnU-Net framework. Patch sampling during training is done randomly and the models we choose for inference are always the final checkpoints after 1000 training epochs. More details are provided in Table 2.

Table 2: Training protocol

| | |
|---|---|
| Network initialization | "he" normal initialization |
| Batch size | 2 |
| Patch size | 40×224×192 |
| Total epochs | 1000 |
| Optimizer | SGD with nesterov momentum ($\mu = 0.99$) |
| Loss function | Dice + CE |
| Initial learning rate (lr) | 0.01 |
| Lr decay schedule | Exponentially decaying (poly lr) |
| Training time | 1000 hours |

## 4   Results and discussion

### 4.1   Quantitative results on validation and test set

The validation DSC scores for various experiments are shown in Table 3. The results from iterations 1-3 were obtained by ensambling the models trained with

5-fold cross-validation, while for iteration 4 only one model was trained using the selected training set. We can make a few observations regarding the use of the unlabelled data. Firstly, using all 2000 cases after training on only the annotated data improves performance by about 2%. Secondly, using uncertainty to select a subset of cases improves performance by an additional 1%. Lastly, continuing this iterative process leads to another 1% increase in performance. No further iterations were performed, because no improvement in performance was observed that warranted spending extra computational resources.

Table 3: Validation DSC per iteration

| Configuration | DSC |
| --- | --- |
| Iteration 0 (50) | 0.8388 |
| Unfiltered iteration 1 (50 + 2000) | 0.8566 |
| Iteration 1 (50 + 401) | 0.8658 |
| Iteration 2 (50 + 1213) | 0.8769 |
| Iteration 3 (50 + 1628) | 0.8761 |
| Iteration 4 (50 + 1813) | 0.8784 |
| Iteration 4 + post-processing | 0.8874 |

The results for each organ in the best performing iteration are shown in Table 4, for both the validation and test set. The worst performing organs were the adrenal glands, followed by the duodenum and the gallbladder. One explanation for this could be that down-sampling to the chosen spacing of $1.2 \times 1.2 \times 4$ mm$^3$ was too aggressive for these small structures, removing necessary information for an accurate segmentation. In general, overall performance was a bit worse on the test set, with mainly the esophagus being segmented less accurately. Interestingly, both the adrenal glands have a slightly higher DSC in the test set.

### 4.2   Qualitative results

Fig. 3 shows the segmentations obtained across different model iterations and Fig. 4 depicts the effect of the post-processing step. From Fig. 3 we can see the clear improvement in segmentation quality for structures like the right kidney (Case 0002), the duodenum (Case 0042) and the left kidney (Case 0006) with the increasing model iterations. These improvements indicate that the model is generating better-quality pseudo-label for the unlabelled portion of the training set at each iteration, which in turn increases the model's generalization power and the performance on the separate validation set. There are however still structures that are incorrectly segmented or not detected at all after iteration 4. We observe that this is more often the case with small structures, due to the aggressive re-sampling performed in the pre-processing step. This can be seen in Fig. 3, where for case 0002 the gallbladder and duodenum are missed

Table 4: Validation and test DSC per organ for Iteration 4 with post-processing

| Organ | Validation DSC | Test DSC |
|---|---|---|
| Liver | 0.9721 | 0.9513 |
| Right Kidney | 0.9374 | 0.9188 |
| Spleen | 0.9565 | 0.9303 |
| Pancreas | 0.8825 | 0.8339 |
| Aorta | 0.9543 | 0.9333 |
| Inferior Vena Cava | 0.8988 | 0.8867 |
| Right Adrenal Gland | 0.7962 | 0.8051 |
| Left Adrenal Gland | 0.7933 | 0.8095 |
| Gallbladder | 0.8265 | 0.8303 |
| Esophagus | 0.8650 | 0.8085 |
| Stomach | 0.9141 | 0.9016 |
| Duodenum | 0.8059 | 0.7656 |
| Left Kidney | 0.9331 | 0.9149 |
| Overall | 0.8874 | 0.8685 |

at every iteration, while large structures such as the liver, stomach and kidneys are accurately segmented. We also observe that the model has difficulties in defining the boundaries of abnormal organs, especially for cases with kidney and and spleen disease. This is depicted in Fig. 3, where for case 0042 a significant portion of the enlarged, diseased right kidney gets classified as liver. One possible reason for these results is that the labelled training set only contains cases with pancreas disease, leading to the generation of noisy pseudo-labels for the cases with other diseases in the training set. Another common mistake we observed was the segmentation of liver lesions as gallbladder. To overcome this issue, at post-processing all liver-enclosed components segmented as gallbladder are re-classified as liver, as is shown in Fig. 4.

To give an impression of the cases that are excluded from new iterations by the uncertainty criterion, a few representative examples are shown in Fig. 5. In the left case, there is a big artifact, which explains the high uncertainty between models. In the right case, all organs are segmented well, except the gallbladder. In this case, it is segmented in two places and far outside the liver. In both these cases the uncertainty criterion works well, because pseudo-label quality is not sufficient for model training. The middle case is an example where there is no clear reason for high uncertainty, but which is still excluded by the criterion.

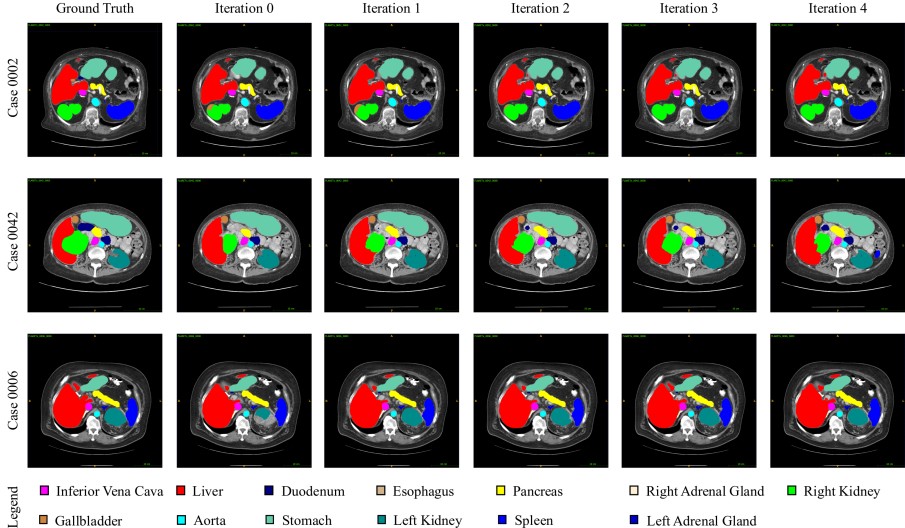

Fig. 3: Segmentation results across model iterations. Clear improvement in segmentation quality are visible for structures like the right kidney (Case 0002), the duodenum (Case 0042) and the left kidney (Case 0006) with the increasing model iterations. However, the model still struggles with the segmentation of small structures (gallbladder and duodenum for Case 0002) and diseased organs (right kidney for Case 0042).

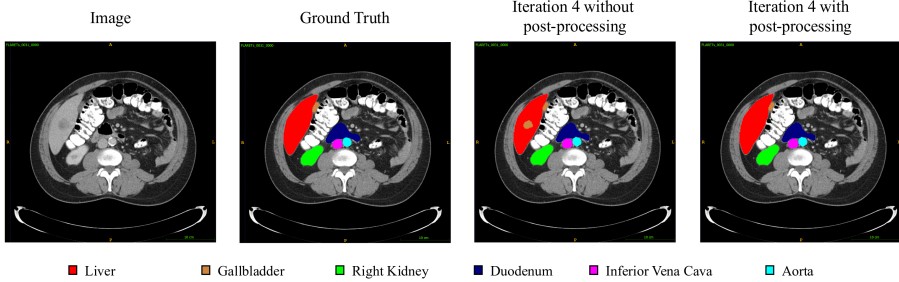

Fig. 4: Effect of post-processing. After iteration 4 the model wrongly segments the liver lesion as gallbladder. This is corrected with the post-processing step.

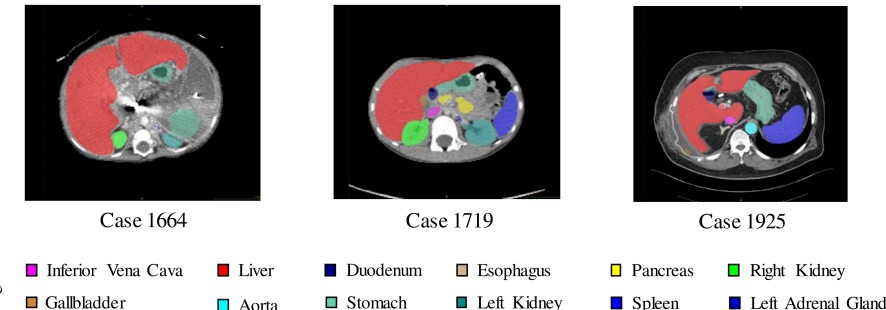

Fig. 5: Examples of cases which did not pass the uncertainty criterion in the final iteration. (left) A case with a severe artifact, (middle) a case with no clear reason for high uncertainty, (right) a case where all organs are segmented well, except the gallbladder, which is segmented far outside the liver.

### 4.3    Segmentation efficiency results

CPU and RAM usage are shown in Fig. 6, for inference on the full validation set of 50 cases. To keep CPU and RAM usage low, inference is limited to a single CPU thread. The GPU memory usage for a single case is shown in Fig. 7 and is approximately 2.7 GB during inference. There is a short peak of 6 GB for the first predicted patch. About 1.3 GB on our testing machine is allocated by NVIDIA drivers, as is visible in the graph at the end of inference. This means that the model, together with the patches it predicts, takes about 1.4 GB.

Different combination of step size and test-time augmentations were tested to minimize GPU time. The ablation results are shown in Table 5. The minimum GPU time was achieved with a step size of 0.9 and no test-time augmentation, with minimal impact on the final segmentation accuracy. This was the selected configuration for the final model. Complete prediction of the validation set of 50 cases takes 22 minutes, meaning that on average the inference time per case is 26.4 seconds.

In the test cohort of 200 cases, the mean time per case was 42 seconds. The difference with the mean time on the validation set could be attributed to a different validation and test environment, or perhaps on average bigger CT volumes in the test set. The area under the GPU memory-time curve and CPU memory-time curve were 7958 and 783, respectively, on the test set.

### 4.4    Limitations and future work

Currently, inference is done with the standard nnU-Net framework. This framework is not optimized for low GPU ($<$2GB) and low RAM ($<$28GB) environments. This may be improved with a custom inference pipeline, for example including ONNX runtime.

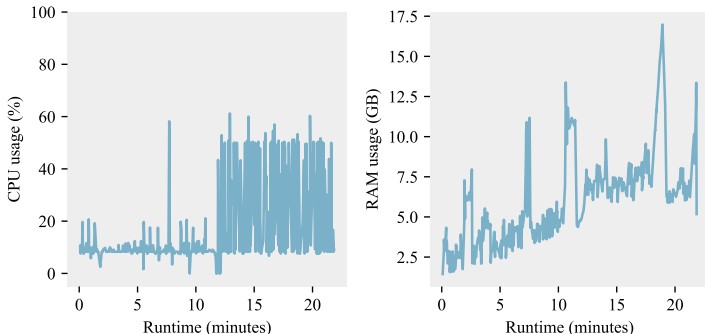

Fig. 6: The CPU utilization (left) and RAM (right) over time for a typical inference session.

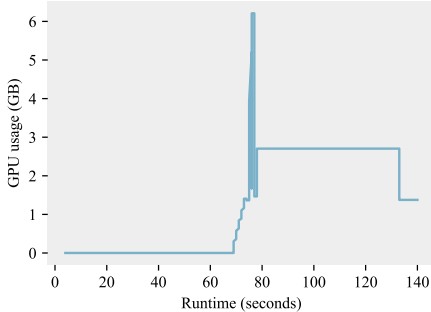

Fig. 7: The GPU memory usage over time a single case.

Table 5: Ablation study for inference step size and test-time augmentation.

| Step size | TTA | DSC | GPU time (s) |
|-----------|-----|--------|--------------|
| 0.9 | no | 0.8848 | 56 |
| 0.9 | yes | 0.8860 | 102 |
| 0.5 | no | 0.8663 | 129 |
| 0.5 | yes | 0.8874 | 284 |

Apart from inference, the training process is very expensive in terms of GPU compute. This is because every iteration requires 5 models to be trained, each consuming roughly 40 GPU hours. The easiest way to improve this is to train each iteration for fewer epochs than the standard 1000 epochs of nnU-Net. Especially for the earlier model iterations it may not be as crucial to train for many epochs, but this has to be explored.

In this paper, we use the mean pairwise DSC per organ as uncertainty metric and propose a threshold of 0.9. For both of these, alternatives are possible, but have not been explored yet. Especially the threshold is interesting to investigate, as it determines the amount of samples per iteration.

## 5   Conclusion

Uncertainty-guided selection of pseudo-labels can improve the< performance of semi-supervised deep learning for multi-organ segmentation on CT. The proposed framework improves segmentation accuracy by 4% compared to not using unlabelled data at all and by 2% compared to using all unlabelled data without selection.

**Acknowledgements** The authors of this paper declare that the segmentation method they implemented for participation in the FLARE 2022 challenge has not used any pre-trained models nor additional datasets other than those provided by the organizers. The proposed solution is fully automatic without any manual intervention.

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
