# OpenReview forum: "Uncertainty-Guided Self-Learning Framework for Semi-Supervised Multi-Organ Segmentation"
_MICCAI.org/2022/Challenge/FLARE_

### Official Review · Reviewer_uQjw · 2022-09-15
**The authors propose an uncertainty-guided self-learning framework for multi-organ segmentation that implements uses psuedo labels to train with unlabelled cases.**

**Rating:** 7
**Confidence:** 3

**Review:**

Strengths: The paper proposes a very interesting technique to measure the uncertainty in the pseudo-label prediction. The authors also make the code publicly available.
Weaknesses: The preprocessing steps should be better described. The model description can be improved.
Details:
- Although the authors mention that they apply the nnU-Net cropping and normalization strategies, these should be described in more detail in the manuscript.
- The text in Fig 1 is very small, increase the font.
- The description of the method can be improved. Include figure that shows the psuedo-labelling process.
- Justify why you use three iterations.

---

> ### Author Response · Authors · 2022-10-10
> **Response review**
>
> 1.Although the authors mention that they apply the nnU-Net cropping and normalization strategies, these should be described in more detail in the manuscript.
> Response: We have added the information regarding cropping and normalization strategies to section 2.1.
> 2.The text in Fig 1 is very small, increase the font.
> Response: We have increased the font in Figure 1.
> 3.The description of the method can be improved. Include figure that shows the psuedo-labelling process.
> Response: We agree with the reviewer that a diagram would help to clarify the proposed approach to leveraging unlabelled that. We have added the diagram in Figure 2.
> 4.Justify why you use three iterations.
> Response: We chose to stop the process after four iterations as we were not observing an improvement in performance that would justify the computational and time resources necessaries to perform more iterations. We agree that this information is not clear in the text and have added it to section 4.1.

---

### Official Review · Reviewer_qof2 · 2022-09-16
**AI that knows when it does not know - Uncertainty-Guided Self-Learning Framework for Semi-Supervised Multi-Organ Segmentation**

**Rating:** 8
**Confidence:** 3

**Review:**

Strengths: The proposed method first trains five models based on labeled data, then generate pseudo labels by the trained models, and finally train one model based on labeled data and selected pseudo-labeled data. The method increases 5% DSC by using selected pseudo-labeled data over using only the labelled data.
Weaknesses:
- In five-fold cross-validation, are training epochs of five models same? How do you determine the training epochs?
- The calculation method of the mean pairwise DSC is not clear. It is better to add a equation.
- Which model is chosen after five-fold cross-validation? Does the method perform better if ensemble the network weights of five models?
- It is better to show some unlabeled samples that are not used for network training.

---

> ### Author Response · Authors · 2022-10-10
> **Response review**
>
> 1.In five-fold cross-validation, are training epochs of five models same? How do you determine the training epochs?
> Response: The model chosen for each fold is the final model after 1000 training iterations, as implemented by default in the nnU-Net framework. The final output for a given case is obtained by ensemble of the 5-folds. We acknowledge that this information may not be clear in the text and so we have reformulated section 2.2 to clarify it.
> 2.The calculation method of the mean pairwise DSC is not clear. It is better to add a equation.
> Response: We agree with the reviewer and have added the equation for the uncertainty estimation in the Methods section.
> 3.Which  model is chosen after five-fold cross-validation? Does the method perform better if ensemble the network weights of five models?
> Response: For three iterations we use ensembling, because this generally gives better pseudo-labels. For the final iteration we train only a single model on all cases that pass the uncertainty criterion, because this is more efficient during inference. This is also mentioned in the text at the end of section 2.2.
> 4.It is better to show some unlabeled samples that are not used for network training
> Response: We agree with the reviewer that this information is interesting and have added it in Figure 5.

---

### Official Review · Reviewer_V6Qt · 2022-09-17
**Uncertainty-Guided Self-Learning Framework**

**Rating:** 8
**Confidence:** 4

**Review:**

The authors propose an uncertainty guided framework for multi-organ segmentation on CT scans that uses a small labelled dataset to leverage a large unlabelled dataset in a semisupervised setting and modifies the nnU-Net framework for efficient inference, e.g., increasing the step size of strided inference from 0.5 to 0.9. The quality, clarity, and description of the paper are good except for some minor issues.

* It is recommended to add a map of uncertainty guided framework for multi-organ segmentation in the method introduction.
* In general, the code link should be at the end of the abstract rather than the introduction.
* It may be more pleasing if table 4 is horizontal and the table header is the abbreviation of the organ.

---

> ### Author Response · Authors · 2022-10-10
> **Response review**
>
> 1.It is recommended to add a map of uncertainty guided framework for multi-organ segmentation in the method introduction.
> Response: We agree with the reviewer that a diagram would help to clarify the proposed approach to leveraging unlabelled that. We have added the diagram in Figure 2.
> 2.In general, the code link should be at the end of the abstract rather than the introduction.
> Response: We agree with the reviewer that this is a more logical place for referencing the GitHub repository, and have adjusted this accordingly in the text.
> 3.It may be more pleasing if table 4 is horizontal and the table header is the abbreviation of the organ.
> Response: We have tried transposing the table, but it becomes too wide. Also, now that we have added the test results to this column as well, we find it better to read in the original form and have therefore kept the original layout.

---

### Official Review · Reviewer_53t5 · 2022-09-18
**The authors propose a progressive training strategy to leverage the unlabeled images of FLARE2022. They trained multiple models using labeled data, and predict pseudo labels for the unlabeled images. Based on the pair-wise DSC, the pseudo labels are selected to train another model.**

**Rating:** 7
**Confidence:** 3

**Review:**

1, The training process is simple, but work for improving the segmentation results.
2, The word "AI" in the title is a large concept, and it is not sutable for the semi-supervised segmentation task in FLARE2022, in my opinon.
3. In Section 2.2, the authors describe the process of training multiple models, including multiple iterations, cross-validation, uncertainy calculation, and seleting the pseudo labels. I suggust the authors provide a diagram here, because it is the key parts of the proposed method. Moreover, The authors should emphasize how to use unlabeled data.
4. The method  has excellent DSC, however, since FLARE2022 challenge is a joint low resource and accuracy oriented segmentation challenge, the authors should better provide the prediction time both in abstract and results as well as DSC.
5. The authors should provide the visualization examples of successful and failed cases.

---

> ### Author Response · Authors · 2022-10-10
> **Response review**
>
> 1.The word "AI" in the title is a large concept, and it is not suitable for the semi-supervised segmentation task in FLARE2022, in my opinion.
> Response: We agree with the reviewer and have changed the title to: Uncertainty-Guided Self-Supervised Learning for Semi-Supervised Multi-Organ Segmentation
>
> 2.In Section 2.2, the authors describe the process of training multiple models, including multiple iterations, cross-validation, uncertainty calculation, and selecting the pseudo labels. I suggest the authors provide a diagram here, because it is the key parts of the proposed method. Moreover, The authors should emphasize how to use unlabeled data.
> Response: We agree with the reviewer that a diagram would help to clarify the proposed approach to leveraging unlabelled that. We have added the diagram in Figure 2.
>
> 3.The method has excellent DSC, however, since FLARE2022 challenge is a joint low resource and accuracy oriented segmentation challenge, the authors should better provide the prediction time both in abstract and results as well as DSC.
> Response: We agree with the reviewer and have added the information about inference time to the abstract and the Results section.
>
> 4.The authors should provide the visualization examples of successful and failed cases.
> Response: These examples are provided in Figure 3.

---

### Official Review · Reviewer_vEsY · 2022-09-21
**Uncertainty-Guided Self-Learning Framework for Semi-Supervised Multi-Organ Segmentation**

**Rating:** 9
**Confidence:** 5

**Review:**

Pros:
- high dice
- proposed techniques are relevant to the task including post-processing for some organs
- overall, authors did a great job

Cons:
- techniques are hardly novel, yet effective

---

### Meta-Review · Program_Chairs · 2022-09-28

**Recommendation:** Minor Revision
**Confidence:** 5

**Metareview:**

Nice paper. Please address the reviewers' comments in the revised manuscript.